# Tourists' Motivation, Place Attachment, Satisfaction and Support Behavior for Festivals in the Migrant Region of China

Yunyao Zhang [1] , Keun-Soo Park [2] and HakJun Song [2],*

[1] Department of Tourism Management, ChongQing Normal University, University Town, Shapingba District, Chongqing 400031, China; zhyunyao@126.com
[2] Department of Hotel & Leisure Management, PaiChai University, 55-40 Baejae-ro (Doma-Dong), Seo-Gu, Daejeon 35345, Korea; kspark5@pcu.ac.kr
* Correspondence: bloodia00@hanmail.net

**Abstract:** This study aimed to explore the relationships among tourist motivation, place attachment, satisfaction and support behavior for hosting festivals in the migrant region of China. A self-administered questionnaire was used to conduct an on-site survey and a second-order structural equation modeling (SEM) technique was employed. The results of the current study showed that visitors' general festival motivations had a positive influence on their place attachment and satisfaction. Visitors' theme-related motivations had a positive influence on their place identity and satisfaction. Place dependence and place identity also positively affected their low-effort support behavior. In addition, visitors' place identity was a positive antecedent of their high-effort support behavior. Visitors' place attachment had a positive influence on their satisfaction and visitors' satisfaction positively affected their support behavior. This study encouraged festival organizers to become aware that place attachment performs an important role in attracting tourists, and nostalgia is one of the most important motivations for hosting festivals in the migrant region of China. As a result, this study provides crucial insights that organizers should pay attention to place attachment and place identity in order to satisfy visitors and support festival activities.

**Keywords:** tourist motivation; place attachment; satisfaction; support behavior; festivals in the migrant region of China; Wushan International Red Leaves Festival

## 1. Introduction

Fraser [1] stated that adjusting to a new environment after migration is a complex process which influences migrants for a long time. By preserving their awareness of cultural belonging within their own networks, migrants tend to successfully shape resettlement, identity, and belonging [2]. In this situation, festivals and special events will be able to contribute to maintaining a sense of cultural belonging for the residents of the region who have migrated as well as the general local residents of a certain region. As festivals can provide opportunities to preserve culture and history in general, migrants can make their customs and traditions appeal to many outsiders by hosting these festivals [3]. Festivals usually celebrate ideologies, identity, community values, and continuity [4]. Heenan [5] asserts that festivals are essentially based on community, and focused on local themes and values, which preserve local culture and show its unique attraction to tourists. Festivals are becoming a culture signal in the construction of today's tourism destinations. Events can be synonymous with a particular place, helping to shape and promote a location, and play a central role in how places are perceived, consumed, and even contested [6]. As an important and distinguished component in the context of leisure and tourism, festivals and special events have received increasing academic consideration in recent decades [7].

Understanding the motivations of festival visitors is a key prerequisite for creating desirable experiences and satisfaction for visitors [7], because it can help managers to improve the position of festivals [8], and create a successful program [9]. Place attachment,

as a sense of physical existence and feeling "in place" or "at home", has been used to describe the phenomenon of human–place bonding [10], which is a mark that an individual has generated an emotional tie to a place [11]. Many studies have found that there is a close relationship between festivals and place attachment [12]. Lee et al. [13] state that there is an imperative mediating effect of place attachment, which plays a significant role to festival destinations in the relationship between festival satisfaction and loyalty. Recently, place attachment has been considered as a useful tool for public land management. In the tourism industry, the understanding of customer satisfaction is also a crucial component for predicting customers' repeat purchasing of products or the re-visiting of tourist destinations [14], in the sense that the estimation and analysis of customer satisfaction can explain customers' future intentions in general [15]. Although prior research has examined the relationships between place attachment, festivals, and tourist motivations, few studies have been conducted to comprehensively explore the relationships among motivation, place attachment, satisfaction and support behavior among tourists in the migrant areas of Eastern nations. Chinese society is now changing rapidly, as outsider cultural invasion has changed the local culture, especially in some migrant areas.

Preserving traditional culture is a complex problem for the migrant areas of Eastern nations. The recognition of tourist-supportive behavior has led to a growing call to promote sustainable festival practices in such settings. If not well managed, increased visitation can put local cultural heritage at risk. High-effort support behavior can bring more benefits to festivals. While high-effort support behavior requires more from tourists, it will help the sustainable development of the festival if the organizers know who is more willing to accept high-effort support behavior and make them more satisfied. In this study, we explain the relationships among motivation, place attachment, satisfaction, and support behavior in festivals, and give some theoretical and practical suggestions for the preservation of traditional culture in those areas. Until now, the migrant region of Three Gorges in China has been not a popular topic in this academic field. Most previous studies have focused on government policy [16], economic and social impacts [17], environmental impacts [18], psychological issues [19], and experiences from Three Gorges [20]. To fill these research gaps, this study aims to develop a theoretical model to examine the structural relationships among tourists' motivation, place attachment, satisfaction, and support behavior of festivals in this migrant region of China. From a theoretical viewpoint, the findings of this study will enrich our understanding of festival tourists' intentions and the processes of participation in festival tours in the migrant areas of China. From a practical viewpoint, this study will contribute by offering festival tourism managers and marketers insights to conduct viable marketing strategies, preserve traditional culture, and attract more festival tourists.

## 2. Literature Review

Iso-Ahola [21] and Crompton and Mckay [22] stated that the construct of motivation can be defined as an internal factor which motivates, directs, and integrates an individual's behaviors. Backman et al. [23] claimed that motivation refers to a driver showing different kinds of behavior toward specific forms of activities, which is related to improving individuals' preferences to arrive at the anticipated satisfactory consequences. A majority of festival and event–motivation studies has been performed under the theoretical framework of travel motivation research [23]. In the first issue of *Festival Management and Event Tourism*, two articles [24] were regarded as "a starting point to understand the motivations people have for attending festivals." Festivals offered a place for families to meet and share a sense of gathering. Different studies identified various dimensions according to the festival themes, and some special events can attract tourists to enjoy their trip [25–28]. Socialization is one of the most important and recurring motivational factors for attendance across previous studies of visitor motivations to attend festivals [25]. Escape is frequently discussed in festival motivation studies [22]. Cultural exploration is one of the important factors explaining visitors' festival motivations [29]. Uysal et al. [30] indicated novelty as the most important motivation of festivals. Excitement and enjoyment are encompassed

by general entertainment [7,25,27,31]. There are many other motivations for joining a festival, including learning (desire to develop skills and techniques as well as to learn about themselves) [32], nostalgia [33], status [7], community pride [31], relaxation [23,34], symbolic meanings [35], and self-expression [36].

Place is a center of meaning constructed by experience. It is not only known by the eyes and mind but also though passive and direct modes of experience, which is objectification [37]. Place attachment is a term that has been used to refer to the emotional bond or connection between people or individuals and specific places [38]. The majority of leisure tourism scholars consistently consider that place attachment includes place identity and place dependence [39,40]. Place identity is defined as "an individual's solid emotional attachment to particular places or settings" [41]. Place dependence is considered as the collection of social and physical resources that meet visitors' specific activity needs and represent the distinctive qualities of a place [38,42].

Satisfaction is one of the most significant factors influencing consumer behavior; therefore, the high level of customer satisfaction is a main concern for all businesses [43]. According to Mason and Paggiaro [43], "satisfaction is a partly affective and cognitive evaluation of consumer's experience". Tourist satisfaction has been widely argued [44, 45], but there is no clear agreement regarding what kind of measurement variables are appropriate. Barsky and Labagh [46] stated that the appraisal of customer satisfaction is one of the most necessary processes to realize business success since it reveals the judgment of a product or service through the customer's response.

Support is a future-oriented action that provides additional feedback [47]. Social exchange theory could be used as an appropriate framework to explain support for tourism development; those who perceived benefit from tourism development usually express positive attitudes and support the tourism [48]. Schroeder [49] stressed that residents' impressions of their state's tourist destinations are closely related to their support for the tourism industry. McGehee and Andereck [50] identified factors predicting rural residents' support of tourism. Nunkoo et al. [51] studied island residents' identities and their support for tourism; the results showed the addition of identity variables in behavioral models could enhance their predictive power in clarifying attitudes to tourism and consequent support for the industry. Nunkoo and Gursoy [52] confirmed the relevance of social exchange theory and identity theory in predicting community support for tourism. Though those studies focused on the residents' perspectives, Song et al.'s [53] research showed that there were positive relationships among festival quality, satisfaction, trust and support from the tourists' perspectives. Lee et al. [13] studied thana and peace tourism from the support behavior perspective. Boyne et al. [54] found that the policy, support and promotion influenced food-related tourism initiatives from a marketing perspective. Ramkissoon et al. [55] found that the pro-environmental behavioral intentions of park visitors can be divided into low and high-effort; satisfaction positively affects low-effort pro-environmental behavioral intentions, and negatively affect high-effort pro-environmental behavioral intentions. High-effort behaviors refer to behaviors that are less obvious and more difficult to perform, while low-effort behaviors are those that take less time and are more obvious to tourists.

Based on multidimensional conceptualizations of motivation and place attachment, Kyle et al. [38] conducted research about the relationship between place preferences and place meaning. The goal of this investigation was to examine the relationship between motivation for interacting with natural settings and attachment to these settings. Hixson et al. [39] aims to identify the event attendance motivation and place attachment. The results showed that motivations for event attendance correlated with place identity, and another variable that affects place identity was found to be the length of residence; these variables have a bearing on the effective bonds that develop between a person and a place, which in this case is the place of residence. Due to the relationship discussed before, hypothetically, Hypotheses 1 and 2 were developed as follows;

**Hypothesis 1.** *Visitors' general festival motivations to attend festivals in the migrant region have a positive influence on their place attachment.*

**Hypothesis 2.** *Visitors' theme-related motivations to attend festivals in the migrant region have a positive influence on their place attachment.*

Motivation, satisfaction and loyalty are repeatedly examined in behavioral studies in various tourism contexts. Empirical studies reported that tourist satisfaction is significantly affected by motivation [36]. Lee and Hsu [36] studied the relationships between motivation, satisfaction and loyalty. The results specify that motivation directly affects satisfaction and indirectly affects loyalty. Due to the relationship discussed above, hypothetically, Hypotheses 3 and 4 were developed as follows:

**Hypothesis 3.** *Visitors' general festival motivations to attend festivals in the migrant region have a positive influence on their satisfaction.*

**Hypothesis 4.** *Visitors' theme-related motivations to attend festivals in the migrant region have a positive influence on their satisfaction.*

According to the literature review, support behavior is introduced in the study as a predictor of behavior, and it has been presented as a direct relationship between value and behavioral intentions [56]. In Western China, for a successful sports hallmark event, support behavior is regarded as one of the most important factors [57]. Support from residents [58] and tourists [13,53] is necessary for a tourism destination. George and George [59] focused their research on the mediating role of place attachment on "frequency and intensity of past purchases" and "intention to revisit". According to the relationship discussed above, Hypothesis 5 was developed as follows:

**Hypothesis 5.** *Visitors' place attachment to festivals in the migrant region has a positive influence on their support behavior.*

The links between such concepts as consumer involvement, service quality, satisfaction, image, motivation, support behavior and loyalty have been frequently confirmed [60]. Valle et al. [61] reported that greater levels of satisfaction resulted in increased likelihood of repeat visit and a powerful willingness to recommend the destination to other people. Jabarin and Damhoureyeh [62] also noted that more satisfaction visitors reported, the more willingness they had to pay for the park. If tourists are satisfied with their travel experiences, they will show affirmative future behavioral intentions, such as the intention to visit destinations again or participate in the same tours again [63]. Due to the relationship discussed above, Hypothesis 6 was developed as follows:

**Hypothesis 6.** *Visitors' satisfaction with festivals in the migrant region has a positive influence on their support behavior.*

A number of scholars reported that the judgment of customer satisfaction could be affected by the form and level of place attachment [64]. The relationship of place attachment and customer satisfaction has been the topic of a number of researchers [64]. Wickham and Kerstetter [65] studied place attachment and its influence on satisfaction with experience and settings. Hou et al. [66] measured visitors' opinions about the diverse components of destination satisfaction and reported that satisfaction with the attractiveness of a tourism destination predicts destination attachment, together with involvement. Due to the relationship discussed above, Hypothesis 7 was developed as followed,

**Hypothesis 7.** *Visitors' place attachment to festivals in the migrant region has a positive influence on their satisfaction.*

## 3. Method

Chongqing is one of the main cities in Southwest China, and is one of the five national central cities in China. The city is also one of China's four direct-controlled municipalities, and a unique municipality in mainland China. Wushan County is located in Chongqing municipality and lies on the northern bank of the Yangtze River, under which in the Three Gorges region was submerged after the construction of the Three Gorges Dam. The old town was uninhibited and flooded under the rising waters, and the new town was constructed on the mountains above. Tourism has played an important role in Wushan, although tourist activity is not as active as it was before the submerging of the Gorges in the first decade of the 21st century [67]. Since 2007, Wushan County has held the Red Leaves Festival every year in November and December, which is known as the best season to enjoy the red maple spectacle. Visitors can enjoy the rich local culture while appreciating the fantastic red leaves along Yangtze River [68]. The 9th Wushan International Red Leaves Festival began on 21 November 2015—during the festival, 426,000 tourists visited the festival and generated USD 619 million in tourism revenue. The 14th Wushan International Red Leaves Festival was held on 1 November to 31 December 2020, as usual, despite of the COVID-19 pandemic situation; fortunately, the festival has not been significantly affected by the pandemic.

To generate research items, a comprehensive literature review on festival visitors' motivation, place attachment, satisfaction, and support behavior was conducted. Motivation was operationalized by 23 items (4 items of Togetherness and Socialization (T&S), 4 items of Escape and Relaxation (E&R), 4 items of Novelty and Excitement (N&E), 4 items of Event Attraction (EA), 3 items of Nostalgia (NT) and 4 items of Cultural Exploration (CE)), which was based on the previous research [9,23,31,32,39,68,69]. Place attachment was operationalized by eight items (four items of Place Dependence and four items of Place Identity). Place Dependence was based on the previous research [11,55]. Place Identity was based on the previous research [40,55]. Satisfaction was operationalized by three items, which was based on the previous research [9,11,55,56,70]. Support Behavior for the Festival was operationalized by eight items (four items of Low-Effort Support Behavior and four items of High-Effort Support Behavior). Low-Effort Support Behavior was based on the previous research [9,11,55,56,70]. High-Effort Support Behavior was based on the previous research [36]. All items were measured using a 5-point Likert scale, ranging from Strongly disagree (1) to Strongly agree (5).

The target population of this research was visitors to the Wushan International Red Leaves Festival in Chongqing, China. The sample was obtained by conveniently selecting participants at the main activity places of Wushan International Red Leaves Festival. Wushan town, Goddess Park, Chaoyun Park, Wenfeng Taoist temple and the port of Goddess River, King of Red Leaves Tree, Little Three Gorges were the main places for the survey. The 9th Wushan International Red Leaves Festival began on 21 November to 31 December 2015—in order to obtain representative samples, the survey was conducted from 17 December 2015 to 3 January 2016. Field researchers illustrated the goal of the research project and invited these tourists to participate in the survey. A self-administered questionnaire was offered to each respondent. Furthermore, the questionnaires were completed in the presence of the field researchers, allowing for rigorous monitoring of the data collection process. The overall response rate of this survey was 93.1% (i.e., 512 completed surveys from the 550 tourists). After an examination, 44 questionnaires were removed from the analysis because some questions were left blank or were checked irregularly. Finally, 468 questionnaires were coded and used for future analysis.

Data collected from the survey were analyzed with SPSS (Statistical Package for the Social Sciences) and the M*plus* program. Data analysis was performed with two processes: preliminary analysis and hypotheses testing. First, SPSS was employed for preliminary analyses such as frequencies, reliability, and exploratory factor analysis. Second, the hypotheses were tested through second order structural equation models using M*plus*. Structural equation model was used with the following two steps: original model testing and extended model testing along with the comparisons of the competing models.

## 4. Results

Table 1 shows the demographic characteristics of the respondents. The proportion of male respondents (56.2%) was greater than that of the female respondents (43.8%). The majority of the respondents were aged 30–39 (30.6%) and 40–49 (29.1%). Tourists from Chongqing (52.6%) were predominant. Most of the tourists travelled with others (71.8%), with the majority in organized groups (32.9%). Most of tourists stayed 6 h–8 h (37.2%). Service and salesperson (16.7%), business manager (15.0%) and farmer (13.2%) were predominant occupations of the tourists.

**Table 1.** Demographic characteristics of respondents (*n* = 468).

| Characteristics | *n* | % | Characteristics | *n* | % |
|---|---|---|---|---|---|
| **Gender** | | | **Occupation** | | |
| male | 263 | 56.2 | Official from government | 39 | 8.3 |
| female | 205 | 43.8 | Business manager | 70 | 15 |
| **Marriage** | | | Technician/academician | 45 | 9.6 |
| single | 96 | 20.5 | Service and salesperson | 78 | 16.7 |
| married | 185 | 39.5 | Worker | 52 | 11.1 |
| divorced or separated | 100 | 21.4 | Military | 9 | 1.9 |
| **Age** | | | Farmer | 62 | 13.2 |
| Younger than 20 | 43 | 9.2 | Retired or no job | 41 | 8.8 |
| 20–29 | 98 | 20.9 | Student | 58 | 12.4 |
| 30–39 | 143 | 30.6 | Other | 14 | 3 |
| 40–49 | 136 | 29.1 | **Education** | | |
| 50–59 | 27 | 5.8 | High school or below | 143 | 30.6 |
| 60 years and older | 21 | 4.5 | College | 105 | 22.4 |
| **Income** | | | University | 124 | 26.5 |
| less than USD 200 | 15 | 3.2 | Postgraduate degree | 62 | 13.2 |
| USD 201–400 | 18 | 3.8 | Missing | 34 | 7.3 |
| USD 401–600 | 50 | 10.7 | **Single traveler** | | |
| USD 601–800 | 46 | 9.8 | Yes | 132 | 28.2 |
| USD 801–1000 | 46 | 9.8 | No | 336 | 71.8 |
| USD 1001–1200 | 87 | 18.6 | **Travel with whom** | | |
| USD 1201–1400 | 55 | 11.8 | Family | 94 | 20.1 |
| more than USD 1401 | 43 | 9.2 | Friend(s)/relative(s) | 75 | 16 |
| **Location** | | | Organized group (school, tour group, work, etc.) | 154 | 32.9 |
| Wushan | 103 | 22 | Others | 13 | 2.8 |
| Chongqing (outside of Wushan) | 246 | 52.6 | | | |
| Outside of Chongqing | 119 | 25.4 | | | |

The most common educational background of respondents is high school or less (30.6%), followed by university (26.5%). Many of the respondents (18.6%) reported their family average monthly income level is USD 1001–1200, USD 1201–1400 (11.8%), or USD 401–600 (10.7%). Most of respondents (39.5%) are married.

The measurement model I (1st order CFA) was employed to examine the construct validity of the first-order factors including eleven exogenous variables (e.g., T&S, E&R, N&E, EA, NT, CE, PI, PD, SAT) and two dependent variables (LSB, HSB). The final structural model showed the following fit indices: $\chi^2$ = 997.412 (df = 724, *p* < 0.000), RMSEA = 0.028, CFI = 0.989, TLI = 0.987, SRMR = 0.027. The results indicate a good fit by exceeding the cutoff criteria suggested by Bearden et al. [71], Baumgartner and Homburg [72], Hu and Bentler [73], and Kelloway [74]. Based on Table 2, all factor loadings were larger than the minimum criteria of 0.5 and the t-values were significantly correlated, which supported the convergent validity of the measurement model for the study model [75]. Reliability and construct validity for the measurement model were examined in Table 3. Due to the

factor loading and $R^2$ of MOV12, the question "I attend the festival to be excited," is 0.446, 0.20, lower than the required, and so removed from the study.

**Table 2.** Results of First-order Confirmatory Factor Analysis for Measurement Model I.

| Factors/Items | Factor Loading | T-Value | $R^2$ |
|---|---|---|---|
| **Factor 1: Togetherness and Socialization (T&S)** | | | |
| I attend the festival because my friends/family want to come here | 0.96 | N/A | 0.91 |
| ________ because I like to meet different people | 0.8 | 25.45 | 0.64 |
| ________ to have the experiences with my friends/family | 0.81 | 27.03 | 0.65 |
| ________ to enjoy the company of the people who came with me | 0.92 | 44.71 | 0.84 |
| **Factor 2: Escape and Relaxation (E&R)** | | | |
| I attend the festival to give my mind a rest | 0.94 | N/A | 0.89 |
| ________ to be free to do whatever I want | 0.84 | 27.15 | 0.7 |
| ________ to escape | 0.82 | 28.57 | 0.67 |
| ________ to change my daily routine | 0.85 | 29.88 | 0.72 |
| **Factor 3: Novelty and Excitement (N&E)** | | | |
| I attend the festival to have fun | 0.97 | N/A | 0.93 |
| ________ to talk about when I get home | 0.98 | 71.66 | 0.96 |
| ________ to enjoy activities that make me thrill | 0.95 | 56.16 | 0.89 |
| **Factor 4: Event attraction (EA)** | | | |
| I attend the festival to see red leaves | 0.99 | N/A | 0.98 |
| ________ to enjoy the activities | 0.93 | 53.93 | 0.87 |
| ________ to see the beautiful scenery | 0.95 | 65.27 | 0.9 |
| ________ to see the film | 0.73 | 19.89 | 0.53 |
| **Factor 5: Nostalgia (NT)** | | | |
| I attend the festival to recall the old town | 0.97 | N/A | 0.94 |
| ________ to reflect on past memories | 0.9 | 40.35 | 0.81 |
| ________ to think about good times in the past | 0.89 | 35.08 | 0.79 |
| ________ to increase my knowledge of migrant culture | 0.93 | 42.67 | 0.86 |
| **Factor 6: Cultural exploration (CE)** | | | |
| I want to experience customs and cultures different from those in my own environment | 0.98 | N/A | 0.96 |
| I want to see the new town | 0.98 | 62.02 | 0.95 |
| I like to find myself in situations where I can explore new things | 0.98 | 62.47 | 0.95 |
| **Factor 7: Place Identity (PI)** | | | |
| Attending this festival means a lot to me | 0.96 | N/A | 0.92 |
| I am very attached to this festival | 0.91 | 36.08 | 0.83 |
| I have a strong sense of identifying with this festival | 0.89 | 41.4 | 0.79 |
| It is a good memory to attend this festival | 0.95 | 52.03 | 0.91 |
| **Factor 8: Place Dependence (PD)** | | | |
| No other festival can compare with this | 0.98 | N/A | 0.97 |
| For me, this festival cannot be substituted by other festival | 0.92 | 51.8 | 0.84 |
| Attending this festival is more important than others | 0.87 | 34.17 | 0.76 |
| I want to stay here more than other festival | 0.93 | 42.66 | 0.86 |
| **Factor 9: Satisfaction (SAT)** | | | |
| I feel very happy with this festival | 0.94 | N/A | 0.88 |
| I am satisfied with my decision to visit this festival | 0.94 | 44.95 | 0.88 |
| Overall, I am satisfied with this festival | 0.99 | 58.65 | 0.99 |
| **Factor 10: Low Support Behavior (LSB)** | | | |
| I will recommend this festival to others | 0.93 | N/A | 0.87 |
| I would like to visit this festival again next time | 0.93 | 34.81 | 0.86 |
| Positive word of mouth to others | 0.94 | 36.14 | 0.88 |
| I will prioritize the festival over other festivals when deciding whether to attend | 0.98 | 41.53 | 0.95 |
| **Factor 11: High Support Behavior (HSB)** | | | |
| Willingness to pay more | 0.97 | N/A | 0.94 |
| Volunteer my time to activities that help this festival | 0.95 | 56.46 | 0.9 |
| I will support this festival as much as I could | 0.92 | 44.35 | 0.85 |
| I will support the effort of Wushan for the development of this festival | 0.92 | 41.62 | 0.84 |

**Table 3.** Results of Inter-Factor Correlations of Measurement Model.

|  | TS | ER | NE | EA | NT | CE | PI | PD | SAT | LSB | HSB |
|---|---|---|---|---|---|---|---|---|---|---|---|
| TS | 1.00 | | | | | | | | | | |
| ER | 0.31 *** | 1.00 | | | | | | | | | |
| NE | 0.47 *** | 0.27 *** | 1.00 | | | | | | | | |
| EA | 0.11 | 0.11 | (0.02) | 1.00 | | | | | | | |
| NT | 0.13 * | 0.24 *** | 0.00 | 0.58 *** | 1.00 | | | | | | |
| CE | 0.57 *** | 0.29 *** | 0.64 *** | (0.10) | (0.09) | 1.00 | | | | | |
| PI | 0.42 *** | 0.46 ** | 0.32 *** | 0.16 * | 0.12 * | 0.36 *** | 1.00 | | | | |
| PD | 0.39 *** | 0.44 *** | 0.36 *** | 0.06 | (0.04) | 0.48 *** | 0.60 ** | 1.00 | | | |
| SAT | 0.49 *** | 0.43 *** | 0.43 *** | 0.13 * | 0.15 * | 0.48 *** | 0.69 *** | 0.62 *** | 1.00 | | |
| LSB | 0.46 *** | 0.45 *** | 0.36 *** | 0.15 * | 0.16 ** | 0.40 *** | 0.71 *** | 0.66 *** | 0.80 *** | 1.00 | |
| HSB | 0.32 *** | 0.36 *** | 0.22 *** | 0.07 | 0.12 * | 0.26 *** | 0.51 *** | 0.42 *** | 0.59 *** | 0.70 *** | 1.00 |
| α | 0.92 | 0.92 | 0.98 | 0.94 | 0.96 | 0.98 | 0.96 | 0.96 | 0.97 | 0.97 | 0.97 |
| CR | 0.93 | 0.92 | 0.98 | 0.95 | 0.99 | 0.95 | 0.96 | 0.96 | 0.97 | 0.97 | 0.97 |
| AVE | 0.76 | 0.74 | 0.93 | 0.82 | 0.96 | 0.85 | 0.86 | 0.86 | 0.91 | 0.89 | 0.88 |

$* p < 0.05.$ $** p < 0.01.$ $*** p < 0.001.$

From the aspect of reliability, each construct has a satisfactory level of reliability due to the values of Cronbach's alpha ranging from 0.92 to 0.98, i.e., larger than the recommended minimum criteria of 0.7 [76]. Convergent and discriminant validity were tested to judge construct validity in Table 3. All AVE (average variance extracted) and composite reliability values for the multi-item scales were larger than the criteria of 0.5 and 0.7, respectively [77]. The results show that this measurement model has a sufficient level of convergent validity.

Discriminant validity was assessed using the correlation between constructs. Fornell and Larcker [78] suggested that AVE was used to check the discriminant validity of constructs in the measurement model; all AVEs of each construct are required to be higher than the squared correlation to demonstrate satisfactory discriminant validity. The confidence interval method is used to evaluate the discriminant validity between two constructs by using a confidence interval of plus or minus two standard errors around the correlation between the constructs and checking whether this interval includes 1.0. If it includes a value of 1.0, the discriminant validity is not confirmed.

The measurement model II (2nd order CFA) was used to explored the construct validity of the second-order factors including eleven exogenous variables (e.g., T&S, E&R, N&E, EA, NT, CE, PI, PD, SAT), two dependent variables (LSB, HSB) and two second order factors (GFMOT, TRMOT). The final structural model showed the following fit indices: $\chi^2 = 1146.673$ (df = 752, $p < 0.000$), RMSEA = 0.033, CFI = 0.984, TLI = 0.982, SRMR = 0.072. The results indicate a good fit by exceeding the cutoff criteria suggested by Bearden et al. [71], Baumgartner and Homburg [72], Hu and Bentler [73], and Kelloway [74]. All factor loadings of the first-order factors were statistically significant ($p < 0.001$), showing that all the second-order factors were well evaluated by the first-order factors. The results are shown in Table 4.

In the structural model (2nd order SEM), SEM was employed to test the structural relationships among MOV, PA, SAT and SB. First, to test the structural model (2nd order SEM), several model fit indices (e.g., chi-square estimate; CFI, TLI, RMSEA and SRMR) were used. Besides the chi-square test of model fit, all other indices support adequate fit ($\chi^2 = 1223.316$; CFI = 0.981; TLI = 0.979; RMSEA = 0.036; SRMR = 0.071). Table 5 indicates the model fit statistics for the first-order latent model, the measurement part of the structural model, and the structural model.

Table 6 and Figure 1 represent the results of the measurement model. In terms of hypothesis 1, visitors' general festival motivations to attend festivals in the migrant region have a positive influence on their place attachment, which was divided into two subhypotheses. In particular, visitors' general festival motivations had positive influences on place dependence ($\beta_{GFMOV \to PD} = 0.67$, t = 13.29, $p < 0.001$) and place identity ($\beta_{GFMOV \to PI} = 0.61$, t = 11.72, $p < 0.001$), supporting Hypotheses 1. Visitors' general festival

motivations to attend festivals in the migrant region have a positive influence on their satisfaction ($\beta_{\text{GFMOV}\rightarrow\text{SAT}}$ = 0.39, t = 5.42, $p$ < 0.001), supporting Hypothesis 3. It seems that visitors' theme-related motivations to attend festivals in the migrant region have a positive influence on their place attachment, which was divided into two subhypotheses. Specifically, visitors' theme-related motivations had a positive influence on their place identity ($\beta_{\text{TRMOV}\rightarrow\text{PI}}$ = 0.19, t = 4.26, $p$ < 0.001), supporting Hypothesis 2a2. However, visitors' theme-related motivations were not statistically significant in predicting desire of place dependence ($\beta_{\text{TRMOV}\rightarrow\text{PD}}$ = 0.04, t = 1.08, not significant), rejecting Hypothesis 2a1. Visitors' theme-related motivations to attend festivals in the migrant region have a positive influence on their satisfaction ($\beta_{\text{TRMOV}\rightarrow\text{SAT}}$ = 0.11, t = 3.11, $p$ < 0.01), supporting Hypothesis 4.

**Table 4.** Results of Second-order Confirmatory Factor Analysis for Measurement Model II.

| Factors/Items | Factor Loading | T-Value | $R^2$ |
|---|---|---|---|
| **Factor 1: General Festival Motivation (GFMOV)** | | | |
| Togetherness and Socialization (TS) | 0.68 | N/A | 0.47 |
| Escape and Relaxation (ER) | 0.43 | 7.68 | 0.18 |
| Novelty and Excitement (NE) | 0.72 | 11.63 | 0.52 |
| Cultural Exploration (CE) | 0.83 | 14.56 | 0.39 |
| **Factor 2: Theme-related Motivation (TRMOV)** | | | |
| Event Attraction (EA) | 0.63 | N/A | 0.85 |
| Nostalgia (NT) | 0.92 | 5.26 | 0.69 |

**Table 5.** Goodness-of-Fit Indices of the Study Model.

| Model | $\chi^2$ | df | CFI | TLI | RMSEA | SRMR |
|---|---|---|---|---|---|---|
| Measurement Model I (1st Order CFA) | 997.412 | 724 | 0.989 | 0.987 | 0.028 | 0.027 |
| Measurement Model II (2nd Order CFA) | 1146.673 | 752 | 0.984 | 0.982 | 0.033 | 0.072 |
| Structural Model (2nd Order SEM) | 1223.316 | 757 | 0.981 | 0.979 | 0.036 | 0.071 |
| Suggested value * | | | >0.9 | >0.9 | <0.08 | <0.08 |

* Suggested values were based on Hair et al. [77], Bearden et al. [71] and Hu and Bentler [73].

**Table 6.** Standardized Parameter Estimates of the Structural Model.

| | Hypotheses | | Coefficients | T-Values | Results |
|---|---|---|---|---|---|
| H1 | H1a1 | GFMOV→PD | 0.67 *** | 13.29 | Accepted |
| | H1a2 | GFMOV→PI | 0.61 *** | 11.72 | Accepted |
| H2 | H2a1 | TRMOV→PD | 0.04 | 1.08 | Rejected |
| | H2a2 | TRMOV→PI | 0.19 *** | 4.26 | Accepted |
| H3 | H3a | GFMOV→SAT | 0.39 *** | 5.42 | Accepted |
| H4 | H4a | TRMOV→SAT | 0.11 ** | 3.11 | Accepted |
| H5 | H5a1 | PD→LSB | 0.21 *** | 7.23 | Accepted |
| | H5a2 | PD→HSB | 0.04 | 1.13 | Rejected |
| | H5b1 | PI→LSB | 0.23 *** | 4.84 | Accepted |
| | H5b2 | PI→HSB | 0.18 *** | 4.14 | Accepted |
| H6 | H6a | SAT→LSB | 0.52 *** | 9.05 | Accepted |
| | H6b | SAT→HSB | 0.44 *** | 7.81 | Accepted |
| H7 | H7a | PD→SAT | 0.15 ** | 3.11 | Accepted |
| | H7b | PI→SAT | 0.35 *** | 7.25 | Accepted |
| Fit Indexes | $\chi^2$ = 1223.316 (df = 757, $p$ < 0.001), RMSEA = 0.036, CFI = 0.981, TLI = 0.979, SRMR = 0.071 | | | | |

** $p$ < 0.01, *** $p$ < 0.001.

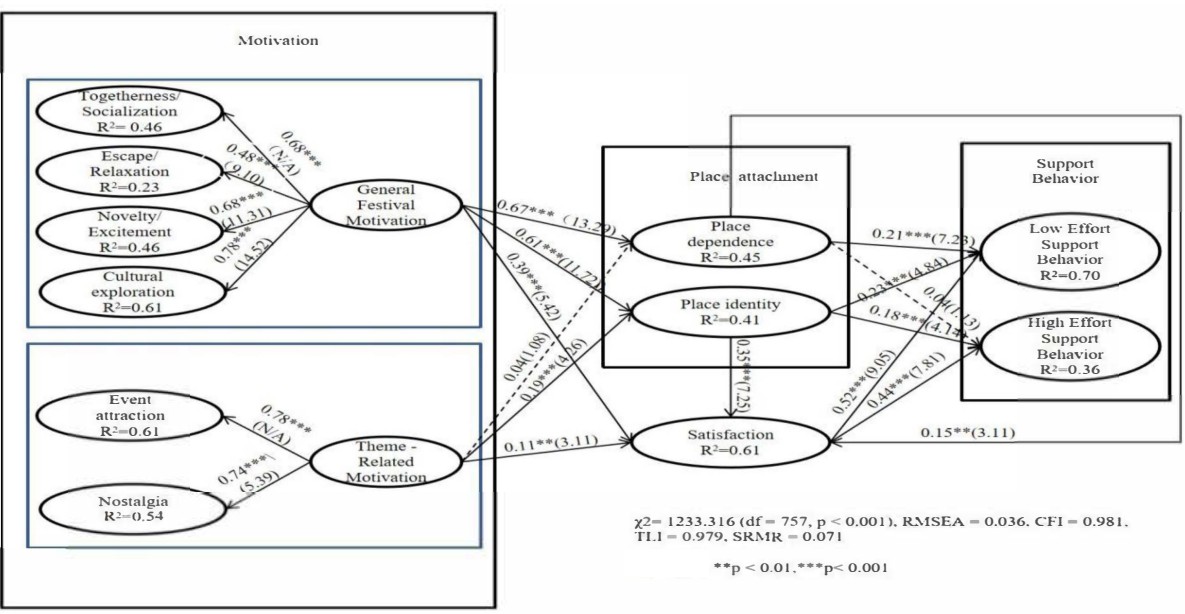

**Figure 1.** Results of the Structural Model.

Visitors' place attachment to festivals in the migrant region has a positive influence on their support behavior, which was divided into four subhypotheses. In particular, visitors' place dependence had a positive influence on their low-support behavior ($\beta_{PD \to LSB}$ = 0.21, t = 7.23, *p* < 0.001) and visitors' place identity had a positive influence on their low-support behavior ($\beta_{PI \to LSB}$ = 0.23, t = 4.84, *p* < 0.001), supporting Hypothesis 5a1 5b1. Although visitors' place identity had a positive influence on their high-support behavior ($\beta_{PI \to HSB}$ = 0.18, t = 4.14, *p* < 0.001), supporting Hypothesis 5b2 is the fact that visitors' place dependence was not statistically significant to predict desire to high-support behavior ($\beta_{PD \to HSB}$ = 0.04, t = 1.13, not significant), rejecting Hypothesis 5a2. Visitors' satisfaction with festivals in the migrant region has a positive influence on their support behavior, which was divided into two subhypotheses. Specifically, visitors' satisfaction was a positive influence on their low-support behavior ($\beta_{SAT \to LSB}$ = 0.52, t = 9.05, *p* < 0.001) and high-support behavior ($\beta_{SAT \to HSB}$ = 0.44, t = 7.81, *p* < 0.001), supporting Hypothesis 6a, 6b. Visitors' place attachment to festivals in the migrant region had a positive influence on their satisfaction, which was divided into two subhypotheses. In particular, place dependence ($\beta_{PD \to SAT}$ = 0.15, t = 3.11, *p* < 0.01) and place identity ($\beta_{PI \to SAT}$ = 0.35, t = 7.25, *p* < 0.001) had a positive influence on their satisfaction, supporting Hypothesis 7a, 7b.

## 5. Discussion and Implications

Using the sample from the Wushan International Red Leaves Festival, the current research attempts to put place attachment into the study of relations between motivation, satisfaction and support behavior related to festivals. Identifying why and how Chinese festival tourists decide to visit festivals in migrant areas is essential for tourism businesses and marketers, because most festival tourism activity takes this form. More specifically, our study analyzed the complex relationships between motivation, place attachment, satisfaction and support behavior in a festival, which suggests that place attachment is an important factor in attracting festival tourists. The results of this study indicate that motivations for attending festivals in the migrant region of China include general festival motivation (GFMOV) and theme-related motivation (TRMOV), which includes togetherness and socialization (T&S), escape and relaxation (E&R), novelty and excitement (N&E), cultural exploration (CE), event attraction (EA) and nostalgia (NT). It was found that the construct of place attachment (PA) can be specifically divided into two subconstructs (i.e., place identity (PI) and place dependence (PD)). The results also indicate that place

attachment theory could be expanded to the tourism motivation, satisfaction and support behavior of festivals in the migrant region of China.

The current study offers several theoretical and practical implications. First, in the literature review and opinions from experts, cultural exploration is regarded as one of the motivations in theme-related motivation, but from the results of the second-order CFA, it is also reasonable that cultural exploration can be regarded as general festival motivation. Cultural exploration is one of the most important factors explaining visitors' festival motivations [29]. Cultural exploration exists in different kinds of festivals, and is likely to be important in an art-oriented festival [22], a world exposition [69], or an international sports event [79], and therefore can be included in general festival motivation. As migrants and refugees often define their current involvement in light of their attachment to their past hometown [80], nostalgia is one of the most important motivations to attend festivals in the migrant region in China, which is consistent with the findings of Ralston and Crompton [33] and Li et al. [81]. The motivation of escape and relaxation (E&R) is the most important factors that leads tourists to participate in festivals in the migrant region of China, which is consistent with the research results of Ralston and Crompton's [33] study. The study found that escape from personal and social pressures is one of the main factors that explained the event-goers' motivations. Socialization is another consistent and recurring motivational factor for attendance across previous studies of festival visitor motivations [25], which is consistent with the results of Crompton and McKay [22] and Schofield and Thompson [79]. Although togetherness is not the first important factor to attract tourists, it is still to be one of the most important factors to attract tourists in this study. Second, place identity is a nostalgic, emotional, or psychological attachment to a place based on a history or built through experiences [82], place identity becomes one of the most important components in place attachment in the study. It shows that place identity is an important symbolic connection between an individual and a setting [83], so for the migrant region of China, place identity became more important than any other factors in place attachment. Place dependence is described as functional attachment to a place based on its importance as a setting for specific activities [42,84]. Place dependence is derived from a transactional view that suggests people evaluate places against alternatives [85]. For this festival, it is just a beginning, not so obvious among alternatives, so place dependence should be enhanced for future development. Although a place has totally changed, the original culture can still be linked to it, indicating that place attachment still plays an important role in the reconstruction of a new society.

Third, the tourists showed low-effort support behavior, with an average value of 4.12, and high-effort support behavior with an average value of 3.02. It reveals that most of tourists prefer to low-effort support behavior than high-effort support behavior, because low-effort support behavior (e.g., recommendation, revisit, positive word-of-mouth) is much easier to do than high-effort support behavior (e.g., willing to pay more, to be a volunteer, support as far as possible). Fourth, in terms of the relationship between motivation, place attachment, satisfaction and support behavior, it was found that motivations have positive impacts on place attachment and satisfaction, and place attachments have positive impacts on support behavior and satisfaction. Satisfaction has positive impacts on support behavior, but there are still two subhypotheses that were rejected. The interesting result was that there was no specific causal relationship between theme-related motivation, place dependence, and high-effort support behavior. The event attraction and nostalgia do not have positive impacts on place dependence, which means theme-related motivation cannot affect place dependence and place dependence cannot promote high-effort support behavior. Understanding the influence factors of place attachment and promoting the support behavior in migrant regions will promote the festival originators to improve the place attachment, satisfaction and support behavior, and thus help the sustainable development of the festival. Place attachment can promote the inheritance of cultural heritage; satisfaction and support behavior can promote the sustainable development of the festival. In

summary, these endeavors could enhance place attachment in the sustainable development of festivals

This study has some limitations and the following steps can provide reference points for future studies. First of all, because we used a convenience sample of Wushan International Red Leaves Festival in Chongqing, the results may not be necessarily generalizable to other migrant regions of China with migrant populations. Future researchers may need to add other variables such as involvement, loyalty and experiences to this model to explain the process better. Finally, although the festival was also held in 2020 without being heavily influenced by the COVID-19 pandemic situation, and visitors who participated in the festival were not significantly affected, there were still limitations due to the fact that the survey for the present study was conducted a relatively long time ago. Therefore, research that can reflect the latest situation needs to be conducted as a future study in order to compensate for these limitations.

## 6. Conclusions

The results indicate that place attachment theory could be expanded to the tourism motivation, satisfaction and support behavior of festivals in the migrant region of China. Place attachment is an important factor in attracting festival tourists, and plays an important role in the reconstruction of the new society. Place identity has become one of the most important components in place attachment in the study, and place dependence should be enhanced for future development. The place dependence suggests that people's evaluation of place was against to alternatives, so they have higher requirements as tourists. Theme-related motivation cannot affect place dependence and place dependence cannot promote high-effort support behavior. Place attachment can promote the inheritance of cultural heritage—satisfaction and support behavior can promote the sustainable development of festival. Cultural exploration can be regarded as general festival motivation, nostalgia is one of the most important motivations, and escape and relaxation (E&R) is the most important factor that leads tourists to participate in festivals in the migrant region of China. Socialization is another consistent and recurring motivational factor for attendance. Although togetherness is not the primary factor in attracting tourists, it is still one of the most important factors in attracting tourists for this study.

**Author Contributions:** Conceptualization, Y.Z. and K.-S.P.; methodology, Y.Z., K.-S.P. and H.S.; software, Y.Z. and H.S.; validation, Y.Z., K.-S.P. and H.S.; formal analysis, Y.Z.; investigation, Zhang, Y.Y; resources, Y.Z., K.-S.P. and H.S.; data curation, Y.Z., K.-S.P. and H.S.; writing—original draft preparation, Y.Z.; writing—review and editing, Y.Z., K.-S.P. and H.S.; visualization, Y.Z., K.-S.P. and H.S.; supervision, K.-S.P. and H.S.; project administration, Y.Z.; funding acquisition, Y.Z. All authors have read and agreed to the published version of the manuscript.

**Funding:** This research was funded by Youth Fund Project of CQU, grant number 12XWQ05.

**Institutional Review Board Statement:** Not applicable.

**Informed Consent Statement:** Informed consent was obtained from all subjects involved in the study.

**Data Availability Statement:** Not applicable.

**Conflicts of Interest:** The authors declare no conflict of interest.

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
