# Peer review of "Tourists’ Motivation, Place Attachment, Satisfaction and Support Behavior for Festivals in the Migrant Region of China"

_sustainability, doi:10.3390/su13095210_

Round 1
Reviewer 1 Report
This was an interesting paper, well-prepared and organized. I would suggest that the author try and come up with a shorter title, something that is catchy that defines the contribution of the study more than the case example itself...this may also help the author(s) with getting the paper cited more in future. The constructs are well-developed, could frame insights on place and events a little more, a recent book may help with this from a conceptual standpoint: Wise and Harris (2019), Events Places and Societies, Routledge, London. Methods section is clear, the results and key findings are highlighted, the discussion offers some scope to lead into the conclusion. I recommend publication after a minor revision and a close edit and proofread of the paper
Author Response
Please refer to an attached file

Reviewer 2 Report
The subject matter presented in the paper is interesting. The authors conducted the study on a large sample of respondents. However, the research was conducted quite a long time ago (from December 17, 2015 to January 3, 2016). Their results, especially now in the context of the COVID-19 pandemic, are out of date.
The aim of the paper has been correctly formulated. The research hypotheses were also properly formulated, although in my opinion there are too many of them. In addition, I would change the numbering of the hypotheses so that the numbers of the hypotheses match the order in which they appear in the article.
In my opinion, a part of the Conclusion should be separated in the work. This part of the paper should present conclusions from the research and references to research hypotheses.
Detailed comments:
- one paragraph sub-chapters in paper should be avoided, such as sub-chapters 2.2, 2.3;
- Results, verse 276. It is: "Most of tourists stay 6-8 hours (18.4%, 18.8%), ...". Why are there two values in bracket? In my opinion there should be one value;
- lines 269-270. What income are we talking about? Monthly? Per capita in the household? Please specify;
- Table 3, why are the correlation coefficients between all the variables not included in the table?
- tables should be formatted and the font should be standardized, etc.,
- Figure 1 in the presented version is difficult to read, please correct it.
Overall, the article is interesting. However, as a result of changes in the world as a result of the COVID-19 pandemic, the study has become obsolete. For this reason, I leave the decision to publish the paper to the editors.
Author Response
Please refer to an attached file

Round 2
Reviewer 2 Report
Thank you for answer for my review.
I think, It is worth add Conclusions to the paper, but I leave it to the Authors' decision.
Author Response
The reviewer's opinion seems to be very valid. Accordingly, a conclusion section has been added to the end of the paper.
Details are as follows.
Results indicate that place attachment theory could be expanded to the tourism motivation, satisfaction and support behavior of festival in migrant region of China. Place attachment is an important factor in attracting festival tourists, and plays an important role in the reconstruction of the new society. Place identity has become one of the most important components in place attachment in the study, and place dependence should be enhanced for future development. The place dependence suggests that people's evaluation of place was against to alternatives, so they have higher requirements on tourists. Theme-related motivation cannot affect place dependence and place dependence cannot promote high effort support behavior. Place attachment can promote the inheritance of cultural heritage, satisfaction and support behavior can promote the sustainable development of festival. Cultural exploration can be regarded as general festival motivation, nostalgia is one of the most important motivations, and escape & relaxation (E&R) is the most important factors that lead tourists to participate in festival in migrant region of China. Socialization is another consistent and recurring motivational factor for attendance. Although togetherness is not the primary factor in attracting tourists, it is still one of the most important factors in attracting tourists for this study.